# Epilepsy and Attention Deficit Hyperactivity Disorder: Connection, Chance, and Challenges

**DOI:** 10.3390/ijms24065270

**Published:** 2023-03-09

**Authors:** Hueng-Chuen Fan, Kuo-Liang Chiang, Kuang-Hsi Chang, Chuan-Mu Chen, Jeng-Dau Tsai

**Affiliations:** 1Department of Pediatrics, Tungs’ Taichung Metroharbor Hospital, Wuchi, Taichung 435, Taiwan; 2Department of Rehabilitation, Jen-Teh Junior College of Medicine, Nursing and Management, Miaoli 356, Taiwan; 3Department of Life Sciences, Agricultural Biotechnology Center, National Chung Hsing University, Taichung 402, Taiwan; 4Department of Pediatric Neurology, Kuang-Tien General Hospital, Taichung 433, Taiwan; 5Department of Nutrition, Hungkuang University, Taichung 433, Taiwan; 6Department of Medical Research, Tungs’ Taichung Metroharbor Hospital, Wuchi, Taichung 435, Taiwan; 7The iEGG and Animal Biotechnology Center, and Rong Hsing Research Center for Translational Medicine, National Chung Hsing University, Taichung 402, Taiwan; 8School of Medicine, Chung Shan Medical University, Taichung 402, Taiwan; 9Department of Pediatrics, Chung Shan Medical University Hospital, Taichung 402, Taiwan

**Keywords:** ADHD, co-morbidity, epilepsy, neurodevelopment, genetics

## Abstract

Comorbidities are common in children with epilepsy, with nearly half of the patients having at least one comorbidity. Attention deficit hyperactivity disorder (ADHD) is a psychiatric disorder characterized by hyperactivity and inattentiveness level disproportional to the child’s developmental stage. The burden of ADHD in children with epilepsy is high and can adversely affect the patients’ clinical outcomes, psychosocial aspects, and quality of life. Several hypotheses were proposed to explain the high burden of ADHD in childhood epilepsy; the well-established bidirectional connection and shared genetic/non-genetic factors between epilepsy and comorbid ADHD largely rule out the possibility of a chance in this association. Stimulants are effective in children with comorbid ADHD, and the current body of evidence supports their safety within the approved dose. Nonetheless, safety data should be further studied in randomized, double-blinded, placebo-controlled trials. Comorbid ADHD is still under-recognized in clinical practice. Early identification and management of comorbid ADHD are crucial to optimize the prognosis and reduce the risk of adverse long-term neurodevelopmental outcomes. The identification of the shared genetic background of epilepsy and ADHD can open the gate for tailoring treatment options for these patients through precision medicine.

## 1. Introduction

Epilepsy is a common neurological disorder affecting 70 million people worldwide [1]. Statistical data show a global lifetime prevalence of epilepsy was 7.6 per 1000 persons; the point prevalence of active epilepsy was 6.38 per 1000 persons, and a peak incidence was in patients aged 20–29 years [2]. Epilepsy/seizure was first described as early as the late Paleolithic period [3] and was mistakenly ascribed to the spells of gods and demons [4]. These supernatural explanations have tortured people with epilepsy to be isolated and feared so far.

With the advent of neuroscience, the definition of epilepsy evolved dramatically due to our better understanding of the disease’s underlying mechanisms and natural history [5]. According to the 2014 International League Against Epilepsy (ILAE) task force, epilepsy is defined as the occurrence of two or more unproved seizures at least one day apart, one unproved seizure in patients with a high risk of recurrence in the next ten years, or presence of epilepsy syndrome [5]. Epilepsy is the most common neurological disorder in the pediatric age group and affects approximately 0.5–1% during childhood [6].

The recent advance in antiseizure medications (ASMs) has expanded largely over the past five decades, including at least 24 different ASMs [7], leading to substantial improvement in the proportion of patients who achieve seizure-free status for three to five years and stop ASMs. However, nearly one-third of patients fail to achieve adequate seizure control [8]. In these patients, childhood epilepsy imposes substantial physical, neurodevelopment, psychological, behavioral, and social burdens; besides, the condition exerts a considerable economic burden that affects the family and public health system [9,10].

Comorbidities are common in children with epilepsy (CWE). Recent reports demonstrated that approximately 50% of people with epilepsy have at least one comorbidity and that the risk of comorbidities may reach eight times higher in people with epilepsy than in the normal population [11]. Broad spectrums of comorbidities can present in CWE, including neurodevelopmental, psychiatric, bone, cardiac, and digestive disorders [11]. Attention deficit hyperactivity disorder (ADHD) is one of the most common psychiatric disorders affecting children. Symptoms of ADHD include impaired attention, impulsivity, and hyperactivity, leading to risk-taking behavior, learning difficulties, disorganization, and difficulty completing tasks [12]. Large population-based studies showed a significantly higher risk of ADHD in CWE than in the general population. The prevalence of ADHD was found to be as high as 77% in CWE, compared to 3–5% in the general population [13]. The presence of ADHD in CWE is challenging and can negatively affect the prognosis of the patients. Previous studies showed that CWE and concurrent ADHD are at increased risk of having an abnormal neurological examination and abnormal background rhythm on the *electroencephalogram* (EEG) [14], lower ASMs adherence [15], and more negative outcomes in social, behavioral, and learning development compared with CWE alone [16].

Despite the high burden, comorbid ADHD remains unrecognized in the clinical setting. Besides, the current literature shows controversies regarding the magnitude and nature of ADHD in CWE, the connection between ADHD and ASMs, the challenges in managing epileptic children with ADHD, and the role of precision medicine in the management of ADHD in CWE. Thus, the current literature review aims to summarize the current burden of ADHD in childhood epilepsy, the possible mechanisms of association between epilepsy and ADHD, and the challenges imposed by ADHD during the management of childhood epilepsy.

## 2. Evolving Concepts in Epilepsy Classification and Etiologies

Epilepsy is a leading cause of disability, morbidity, and economic burden worldwide. Current data indicate a global age-standardized prevalence of epilepsy of 540.1–737 per 100,000 population [17]. Nevertheless, the burden of epilepsy shows notable demographic and socioeconomic disparities, with a striking difference in its incidence between developed (33–82/100,000) [18] and developing (up to 187/100,000) countries [18,19]. The prevalence of epilepsy was reported to range from 1.5 to 40 per 1000 population [9]. In the pediatric age group, the estimated prevalence was 6.8 per 1000 in the USA [20], 3.2–5.1 per 1000 in Europe [21], 1.5 to 14 per 1000 in Asia [22], and 3.3 per 1000 in Taiwan [23]. The incidence of epilepsy is highest in the first year of age and decreases to adult levels by the end of 10 years of age [24]. Prevalence and incidence of epilepsy are relatively higher in males than in females [2].

CWE shows a substantial diversity in the seizure onset, type, duration, associated syndromes, and underlying etiologies. Therefore, epilepsy classification acts as a framework for definitive cost-effective diagnosis and optimal management of childhood epilepsy [25]. The subjective symptoms and objective characteristics (collectively termed semiology) play a central role in diagnosing and managing epilepsy. Therefore, with the aid of video EEG, the revised ILAE classification of epilepsy in 1989 put more emphasis on epilepsy semiology, and the classification was based mainly on seizure characteristics and consciousness level [26]. However, with the advances in our understanding of cellular mechanisms and wide use of invasive monitoring, such as stereo EEG (SEEG), the ILAE classification underwent several revisions for a more comprehensive multi-level classification incorporating ictal semiology, seizure types and localization, and degree of conscious impairment [27]. Nevertheless, these revisions were criticized for including unnecessary terminologies and complexity, which limited their applications in routine practice and continued use of the 1989 classification [28].

Thus, with the evolving evidence of a strong correlation between the semiologic features and SEEG-recorded activated cortical zones during seizure episodes [29], the ILAE published a new classification in 2017 that not only heavily relies on the semiologic features but also incorporates etiology and the risks of comorbidities (Figure 1) [30]. The 2017 multi-level classification begins with the clinical identification of seizure type, which is now operationally classified into focal, generalized, or unknown onsets [31]. The proportion of generalized and unknown seizures peaks in children less than one year and decreases with age, while the proportion of focal seizures shows its highest peak in children aged 5–9 years [32]. In the second classification level, generalized seizures are categorized according to the presence of motor onset, while focal seizures are classified according to the associated awareness impairment and the presence of motor or non-motor symptoms. In some instances, generalized and focal seizures co-exist in EEG and clinical presentation, such as Dravet syndrome. The third level in the 2017 ILAE classification is to identify epilepsy syndrome, referring to a distinct entity characterized by the presence of aggregate clinical, EEG, and imaging features. Besides, some epilepsy syndromes incorporate age at onset, prognosis, diurnal variations, and/or comorbidities [30]. Epilepsy syndromes can play a significant role in predicting treatment response and prognosis of the patients; certain syndromes—such as West syndrome and Dravet syndrome- have a well-established association with drug resistance, cognitive impairment, and psychosocial outcomes [30]. Importantly, several clinical trials showed a syndrome-specific efficacy of traditional ASMs and hormonal therapy [33,34].

Nearly 75% of patients were historically classified as having epilepsy of unknown etiology. However, with the advancement in gene-sequencing studies and neuroimaging, there was a paradigm shift in identifying epilepsy etiologies, and several genetic mutations were incorporated into the etiology of epilepsy [35]. A better understanding of the underlying etiology can open the gate for the clinical application of precision medicine in the epilepsy setting and the development of targeted therapies that tackle the neurobiological pathways for epilepsy [36]. Therefore, the 2017 ILAE classification emphasizes identifying the underlying etiology of epilepsy, whenever possible, to guide clinical judgment and management algorithms. Etiologies for epilepsy are broadly classified into six categories (see Figure 1), with overlapping causes in many patients.

Congenital or acquired structural damages are common causes of childhood epilepsy, and their identification is crucial in achieving seizure-free status through appropriate interventions, such as respective surgery. On the other hand, metabolic conditions—such as mitochondrial cytopathies- can lead to epilepsy and require specific metabolic therapy to optimize patients’ outcomes. Autoimmune etiologies have also been implicated in epilepsy secondary to the development of autoantibodies leading to inflammatory changes and encephalitis; autoimmune epilepsy usually responds to immunomodulatory therapies. Infectious etiology was also identified in patients with epilepsy, such as tuberculosis and meningococcal meningitis. Epilepsy of unknown etiology implies no identified causes and is usually associated with normal imaging findings and a lack of relevant history.

Several epilepsy syndromes are classified under genetic etiology—such as generalized genetic epilepsy syndromes-; nonetheless, it should be noted that a genetic etiology is not synonymous with generalized epilepsy; many focal epilepsies have known monogenic or complex polygenic causes [35]. De novo mutagenesis is increasingly identified in cases with epilepsy, donating a clear distinction between “genetic” and “inherited” epilepsy, as well as explaining the limited familial liability of epilepsy [37]. Certain genetic mutations were also linked to developmental delay and poor prognosis in CWE.

Lastly, the 2017 ILAE classification identified comorbidities as integral components during the evaluation and management of epilepsy due to their profound impact on the patient’s outcomes and quality of life. In the next section, we discuss the burden of comorbidities in CWE.

## 3. The Burden of Psychiatric Comorbidities in Epilepsy

The burden of comorbidities is high in patients with epilepsy; compared to the general population [11]. The burden of comorbidities in patients with epilepsy also demonstrates distinct disparities between high and low-income countries and between the adult and pediatric populations. For instance, the comorbidities in patients (including infants and adults) with epilepsy were: depression (13%), psychosis (10.4%), alcohol abuse (8.7%), and drug abuse (7.8%) in the United States [38]. Comparably, in a systematic review and meta-analysis that covered 39 low and middle-income countries, the distribution of comorbidities of patients (including children and adults) with epilepsy in the published reports was as follows: infectious diseases—such as neurocysticercosis and meningitis–(44%), somatic comorbidities -such as cranial trauma, malnutrition, stroke, and diabetes- (37%), psychosocial (11%), and psychiatric comorbidities (8%) [39]. In adults, approximately half of patients with active epilepsy have at least one comorbidity [11,40]. Reported comorbidities include mental and behavioral disorders and diseases related to the endocrine system, nervous system, circulatory system, respiratory system, digestive system, genitourinary system, musculoskeletal system, connective tissues, etc. [11]. In the pediatric population, nationwide reports indicated that up to 80% of CWE suffer from at least one comorbidity, with a notably high burden even in cases with complicated epilepsies [41,42]. In a population-based in the United Kingdom that included 12,720 patients with epilepsy, nearly 70% of patients had one or more comorbidities, and 30.3% had three or more comorbidities [43]. Several comorbidities were reported in patients with epilepsy, including infectious diseases, neurological disorders (such as stroke, cerebral palsy, and malformations), non-neurological malformations, peptic ulcers, osteoporosis, neurodevelopmental, and psychiatric comorbidities [44]. Comorbidities can negatively impact the long-term developmental, learning, and psychological outcomes of affected children, as well as the quality of life of the patients and their families [44,45].

Psychiatric comorbidities are prevalent in CWE, and the association between epilepsy and a higher risk of psychiatric disorders is well-established [46]. Data from the United States (US) showed that 39.9% of CWEs had at least one psychiatric disorder [38]. Another nationwide study from the US on six million children showed that the risk of neurodevelopmental disorders was significantly higher in children with newly diagnosed epilepsy (60% versus 23%) [47]. Data from the UK also show that the prevalence of psychiatric comorbidities ranges from 20–30% [48]. In two reports from Norway, the prevalence of psychiatric comorbidities in CWE ranged from 37.7% to 42.9%, compared to 6.6% in children from the general population [42,49]. In India, nearly one-third of CWE were found to have psychiatric comorbidities [50]. Notably, in the abovementioned registries, neurodevelopmental and behavioral disorders were found to take a considerable pole in CWE and psychiatric comorbidities. In the Norwegian registries, 21.3% of the children were found to have neurodevelopmental disorders [42]. Jason et al. collected data from the Swedish Registry of Epilepsy and found that 32% of children had neurodevelopmental comorbidities at diagnosis, which increased to 35% after two years [51]. In 2012, Russ et al. utilized the 2007 National Survey of Children’s Health data and found that CWE were more likely to have neurodevelopmental disorders than the general population [52]. In CWE, the most common neurodevelopmental disorders include neurodevelopmental delay, behavioral and emotional disorders, autistic spectrum disorder, intellectual disability, and ADHD [44].

## 4. Evolving Concepts in ADHD

ADHD is a neurodevelopmental spectrum disorder affecting mainly preschool and younger children and is characterized by excessive activities and impaired attention and concentration. In return, children with ADHD usually exhibit disorganization, difficulty in completing tasks, learning disabilities, impaired decision-making, emotional dysregulation, and risk-taking behavior [12]. It is the most common behavioral disorder in children, with nearly 5–7% of the global pediatric population having ADHD. The distribution of ADHD shows a notable geographical disparity. Recent reports showed an increase in the number of children with clinically diagnosed ADHD in the past few decades [53]. A report from Sweden showed a fivefold increase in the number of children with a clinical diagnosis of ADHD from 2004 to 2014, which is mainly attributed to increased awareness of ADHD among general pediatricians and the public rather than an actual increase in the number of children with diagnostic-level of ADHD [54].

Although the precise pathogenic mechanisms underpinning the development of ADHD are unclear, however, ADHD exhibits functional and structural abnormalities. Previous reports suggested the involvement of dysregulated neurotransmitters, including dopamine and norepinephrine [55], reduced cerebral blood flow in the prefrontal region [56], reduced gray matter volume in the prefrontal, parietal, striatal, and cerebellar areas [57] and in the basal ganglia [58], and dysfunction in the right inferior frontal cortex, striatal-thalamic areas, anterior cingulate cortex, and supplementary motor area [59,60,61], in the pathogenesis of ADHD.

The current literature describes several environmental and genetic risk factors for the development of ADHD in children. Environmental factors related to maternal parts, such as advanced age, socioeconomic factors, and psychosocial deprivation [62], smoking [63], alcohol [64], lead [65], polychlorinated biphenyls and dioxins [66] and anemia; neonatal parts, such as September births, low birth weight, breech delivery, prematurity, hypoxic-ischemic encephalopathy, small head circumference, cocaine and alcohol exposure, and iodine and thyroid deficiency [67]; low levels of Mg, Fe, Zn, Cu and Se in the children with ADHD [68]; environmental factors: air pollution [69], and infection diseases, such as influenza, human immunodeficiency virus, *Enterovirus 71*, Varicella Zoster, *Borrelia burgdorferi*, streptococcus, acute otitis media [67], showed strong associations with ADHD. Genetic factors represent the most significant predictors of ADHD, according to genome-wide association studies (GWAS) and nationwide observational studies. According to a 2010 meta-analysis by Nikolas et al., it was estimated that the heritability of ADHD is as high as 70–80% after controlling for potential confounders, such as sex, age, and diagnostic approach [70]. Besides, twin studies demonstrated a strong correlation between genetic factors and clinical traits of ADHD, donating that genetic factors also influence symptom distribution of ADHD [71]. ADHD is primarily a polygenic disorder; recent GWAS identified 304 genetic variants in 12 independent loci associated with the risk of ADHD [72]. Single-nucleotide polymorphisms (SNPs)-based heritability can present in up to 34% of children with ADHD, with a significant impact of these SNPs on symptoms score, suggesting both a polygenic architecture of ADHD and genetically influenced phenotype [73].

Clinically, the diagnosis of ADHD is challenging because the diagnosis of this disease only can rely on the established diagnostic criteria, which include observing specific behaviors in multiple settings. However, errors may not be avoided because of substantial variations in presenting symptoms and extent of impairment, and the subjectivity of the observers. Moreover, the definition and diagnosis of ADHD have evolved since its recognition as a functional disturbance rather than so-called “minimal brain dysfunction” [74]. Initially, the condition was called “hyperkinetic reaction of childhood disorder” in the second edition of the Diagnostic and Statistical Manual of Mental Disorders (DSM-II), implying symptoms of excessive motor activity. Later, the disease was termed “attention-deficit disorder” in the DSM-III. However, with the progress in understanding the potential causes and various presentations, the condition was then termed ADHD in the subsequent DSM editions [75]. According to the DSM-V, children up to 16 years old should have at least six symptoms in the two domains of ADHD (hyperactivity/impulsivity and intention) in at least two separate settings. Adults and adolescents should have at least five symptoms in each of both domains (Table 1). In both age groups, the symptoms should be disproportional to the developmental stage of the patients and persist for six months [76].

Several changes were made in the DSM-V regarding the diagnosis of ADHD compared to previous editions. The age at onset increased from seven to 12 years old to account for the fact that significant ADHD-related impairments can persist in adolescents and younger adults [77]. Besides, ADHD no longer belongs to disruptive behavior disorders, allowing for a separate diagnosis of ADHD and autism. The DSM-V also changed the term “subtype” to “presentation” to reflect that ADHD manifestations are dynamic and can change in adulthood. According to the predominant symptomatology, the DSM-V classifies ADHD into three presentations: primarily inattentive, primarily hyperactive, and a combined type [76].

## 5. The Burden of ADHD in CWE

The association between ADHD and CWE is under-recognized in clinical practice [42,43,78]. Statistical data shows that the reported prevalence of ADHD in CWE ranges from 12–70%, which is primarily attributed to disparity in screening tools, diagnostic criteria, and studied age groups [13,16,42,49,52,79,80,81,82,83,84] (Table 2). However, the prevalence of ADHD in healthy children without epilepsy is approximately 5–15% [42]. Obviously, CWE has a higher risk of ADHD than the general population.

Primarily inattentive presentation of ADHD appears to be more common than other presentations in CWE, which contradicts the distribution of ADHD presentations in the general population [85]. Previous reports showed that problems with attention were more common in ADHD patients with epilepsy than in children with ADHD alone [86]. However, other reports showed a similar distribution of ADHD presentations in children with ADHD and epilepsy and the general ADHD population [87].

Several risk factors were implicated in the co-presence of ADHD in CWE. Although ADHD is more common in boys than girls, reports on CWE showed equal sex distribution of ADHD [85]. Besides, case-control studies showed that the male gender was not predictive of ADHD in CWE [49,88]. The literature also indicates conflicting results regarding the association between age at the onset of epilepsy and ADHD. Further evidence is required to establish the association between younger age at seizure onset and ADHD [89,90]. Besides, environmental factors, such as prenatal exposure to ASMs, increased the risks of ADHD symptoms in CWE [91].

On the other hand, previous reports demonstrated a trend toward an association between seizure characteristics and the risk of ADHD in CWE. For example, two reports showed a positive correlation between impaired level of attention and temporal lobe epilepsy or childhood absence epilepsy (CAE) [92,93]; however, it should be noted that the other reports showed no correlation between seizure type and ADHD, reflecting that the current evidence is controversial regarding the association between seizure type and ADHD [85,88]. Single-center experience also reported a significant association between increased spike index or interictal spikes and impaired attention [89,94,95]. However, the ILAE consensus on ADHD in CWE concluded that the current evidence is still limited to support the association between EEG variables and the risk of ADHD [91]. Seizure frequency has also been linked with the risk of ADHD in CWE; prospective studies showed that, in children with drug-resistant epilepsy, higher seizure frequency was a significant predictor of the presence of ADHD symptoms [96,97]. Limited evidence is available regarding the association between ASMs and the development of ADHD. ASMs, such as phenobarbital, phenytoin, and valproic acid, were found to increase the risk of cognitive impairments, including inattention and hyperactivity [98]. However, it is still unclear whether there is an association between polypharmacy and the risk of ADHD, which requires future studies to investigate whether the adverse effects of polypharmacy increase the risk of ADHD comorbidity in CWE syndromes. Family history can play a role in the development of ADHD in CWE. A Previous single-center report showed that CWE and comorbid ADHD had a significantly higher rate of maternal history of ADHD than CWE only [99]. Nevertheless, there is a strong link between ADHD and epilepsy. Moreover, the impact of comorbid ADHD in epileptic patients is heavily, often associated with academic and occupational underachievement, anxiety, and depression [100].

## 6. ADHD and Epilepsy: Mechanisms of Association

Although a myriad of evidence indicates the high burden of ADHD in CWE, the precise mechanisms underpinning the association between ADHD and epilepsy are unclear. Several models were proposed to explain the reasons for the association between epilepsy and comorbidities, which are usually interchangeable and can co-exist in the same individuals. In 2009, Kaufmann et al. [101] proposed three hypotheses for the possible mechanisms of association between epilepsy and comorbidities. In 2016, Keezer et al. [11] extended these hypotheses and incorporated several aspects of other theories to classify the potential mechanisms of associations into five main categories. In this review, we adopted this framework to highlight the possible association mechanisms between ADHD and epilepsy.

### 6.1. ADHD and Epilepsy: A Chance?

ADHD and epilepsy are common in children; thus, it is expected to observe the co-existence of both disorders in some children. However, mounting evidence as suggested by Chou et al. [102] that ADHD may significantly increase the risk of epilepsy developing and vice versa. The magnitude of the association between epilepsy and ADHD largely rules out the possibility of a chance in this association.

### 6.2. Causative Mechanisms

The relationship between epilepsy and ADHD can be divided into direct and indirect based on the causative mechanisms. For example, ADHD may directly be attributed to the temporal effects of epileptic discharges [103] or may indirectly be due to the side effects of ASMs [104].

### 6.3. Resultants Mechanisms

CWE and ADHD patients exhibited significant adrenergic system dysfunction, which can explain the association between both disorders [105]. Previous experiments demonstrated that rats with induced epilepsy and interictal spikes showed significant levels of ADHD-related symptoms, such as inattention and impulsivity, which were attributed to reduced noradrenergic transmission [106,107]. Thus, it can be hypothesized that adrenergic system dysfunction in CWE acts as a resultant mechanism for comorbid ADHD.

On the other hand, the current body of evidence showed an association between comorbid ADHD and complicated epilepsy [92,93], increased spike index [89,94,95], and higher seizure frequency [96,97]. Thus, researchers have also suggested that neuronal damage in children with complex or uncontrolled seizures may contribute to the increased risk of ADHD in these patients.

### 6.4. Shared Genetics

The past few decades have witnessed dramatic advances in our understanding of genetics’ role in the development of both epilepsy and ADHD, even though these are seemingly two distinct diseases but are with a common genetic background. For instance, some chromosomal abnormalities, such as 5q14.3 and 7p22.3, were found in both ADHD and epilepsy [108,109]. Likewise, several single nucleotide polymorphisms (SNPs), including myocyte enhancer factor 2 (MEF2C)-related disorders and methyl-CpG binding protein 2 (MECP2), were also found to have a high rate in children with ADHD and CWE [110]. A growing body of genetic studies suggested a significant familial liability for epilepsy and comorbid ADHD in children with a history of maternal ADHD [99] and epilepsy [111], suggesting familial clusters of that ADHD and epilepsy. Relatedly, Brikell et al. demonstrated that the risk of comorbid ADHD increased significantly in children with a family history of epilepsy; genetic correlation explained 40% of the phenotypic correlation between epilepsy and ADHD. Notably, the risk was higher among full siblings than half-siblings, further confirming the familial genetic liability [112], suggesting that the two diseases may have overlapped genetic pathogenesis.

In return, biological models were developed to explain the genetic overlap between epilepsy and ADHD. ADHD is a neurobiological disorder that encompasses several biological dimensions with multifactorial interactions. Thus, ADHD and epilepsy may share certain aspects of biological dimensions, despite the distinct clinical phenotypes [110]. Therefore, comprehensive omics data analysis is needed to investigate the biological overlap between ADHD and epilepsy. Another proposed explanation for the genetic overlap between epilepsy and ADHD is the fact that ADHD is a spectrum disorder, and epileptic seizures can be defined along a single spectrum.

### 6.5. Shared Risk Factors

As elaborated by Brikell et al., the genetic correlation explained only 40% of the phenotypic correlation between epilepsy and ADHD [112], which is lower than the findings of the correlation between ADHD and other neurodevelopment disorders [113]. Such results suggest that the genetic correlation between ADHD and epilepsy does not fully explain the high burden of comorbid ADHD in epilepsy and that environmental or other factors play a significant role in this association [114]. Environmental factors, such as toxins, low birth weight, advanced maternal age, socioeconomic factors, and psychosocial deprivation, showed strong associations with ADHD risk in well-controlled studies [62]. Recently, a study merging the daily air pollution data from the Environmental Protection Administration and the population data including children with ADHD from the Taiwan National Health Insurance Research Database concluded that exposure to higher levels of air pollutants in early life, the risk of developing ADHD was higher [69]. Consistently, these factors were also implicated in the development of epilepsy [115,116,117,118,119].

### 6.6. Bidirectional Connection

Bidirectional effects imply that co-existing conditions can cause each other. Bidirectionality can be assumed when a reciprocal temporal sequence is established. Previous reports showed that autism spectrum disorder, another neurodevelopmental disorder, had a bidirectional connection with epilepsy and that children with autism have a higher risk of developing epilepsy than the general population [120]. As previously stated, the prevalence of ADHD is higher in CWE than in the general population. CWE has two-to-seven times higher risk of ADHD than the general population [42,44,78]. Notably, Liu et al. found that ADHD symptoms were more prevalent in new-onset epilepsy than in healthy controls (31% versus 6%), reflecting that ADHD occurred in these patients before epilepsy onset [121]. The bidirectional connection between epilepsy and ADHD was established by a population-based study that included children with new-onset epilepsy and new-onset ADHD from the Taiwan National Health Insurance Research Database. The results revealed a 2.54-fold increased risk of subsequent ADHD in CWE compared to controls. At the same time, children with new-onset ADHD had a 3.94-fold increased risk of subsequent epilepsy [102]. These findings suggest that clinical or subclinical epileptiform discharges are likely to involve the onset of ADHD, especially attention problems. Therefore, these two conditions substantially have a bidirectional connection, and even mutually overlap to some extent the bidirectionality between epilepsy and ADHD can be viewed in the light of the shared genetic factors, pathological changes, and environmental risk factors rather than as evidence that ADHD can lead to epilepsy or vice versa.

## 7. ADHD in Different Types of Epilepsy

The prevalence of ADHD is significantly higher in CWE than in normal controls (Table 2). Because it is not clear whether the cause for the connection between ADHD and CWE is due to a global attention deficit or a specific network deficit, therefore, the connection between ADHD and CWE has been presumed to be attributed to interactions between ongoing seizures, underlying causes of symptomatic epilepsy or innate properties of epilepsy [122,123,124,125]. There are several types of epilepsy associated with ADHD discussed in the following.

### 7.1. Generalized Epilepsy

Generalized seizures start when all brain areas are affected by an abnormal, widespread, excessive electrical impulse. Symptoms include losing muscle tone, clonic, jerking, or stiffening limbs, staring and blinking, and rhythmic, full-body jerking, typically provoked by sleep deprivation, fatigue, and alcohol drinking [126]. Antecedent and birth history, neurological examination, intelligence, and head size are typically normal. Most children with seizures were found to be in a generalized manner [127]. Attentional difficulties are more common among patients with generalized epilepsies than those with partial seizures [85,128]. Some of them also show memory impairment, learning problems, and psychomotor impairments [129]. The neurological basis for such cognitive impairments in patients with generalized epilepsy is unclear. Impaired brain development in children with epilepsy has been linked to a range of cognitive and behavioral issues, including the development of ADHD.

### 7.2. Frontal Lobe Epilepsy (FLE)

The frontal lobes, which constitute almost one-third of the human brain, are the largest parts of the brain. Frontal lobes can incorporate with other parts of the brain and contribute to overall brain function. FLE is the second most common partial epilepsies in childhood [130]. Any epileptic discharges within the frontal regions may potentially affect frontal lobe-associated functions, such as motor functions, control of continence, and olfaction, voluntary eye movements, speech and language abilities, executive functions, motivational behaviors, and social competency. Impairments of those functions may lead to variable symptoms in patients with FLE [131]. The manifestations of FLE include nausea, vomiting, vertigo, motor and cognitive abnormalities, abnormal body posturing, tics, disinhibition, excessive excitement/irritability, inattention, and impulsivity [132]. The psychiatric manifestations of FLE may be difficult to differentiate from those with ADHD without epilepsy, and ADHD may mimic FLE with or without ADHD and vice versa [133]. As the underlying mechanisms for FLE are that frontal cortex abnormalities dysregulate the frontal lobe networks, leading to attention and hyperactivity at a crucial early stage of brain maturation [134], it may be easily confused with ADHD [132]. Nocturnal FLE, is defined by seizures that attack in sleep and originate in the orbitofrontal or mesial frontal areas, and has been reported to be associated with ADHD [135]. Patients with tuberous sclerosis involving frontal lobes were reported to present seizures with concomitant ADHD [136].

### 7.3. Childhood Absence Epilepsy (CAE)

CAE is one of the common forms of pediatric epilepsy. Approximately 10–17% of all childhood-onset seizures are determined to be typical CAE [137], which presents with a short duration (4 to 30 s), transient disturbance of consciousness, blinking, staring, and/or subtle or motionless activity without post-ictal symptoms, and less commonly, various types of automatisms, occurring numerous times per day and happen without being noticed for long. CAE is typically provoked by hyperventilation and sleep deprivation. EEG shows generalized 2.5 to 5 Herz spike wave activity [138]. A study reports that 25% of CAE children have subtle cognitive deficits and ADHD is significantly associated with CAE (26% of children with CAE vs. 6% of normal children) [139], suggesting cognitive or psychiatric comorbidities, especially ADHD are associated with CAE.

### 7.4. Benign Rolandic Epilepsy (BRE)

BRE, or benign epilepsy with centrotemporal spikes (BECTS), is a common idiopathic focal epilepsy in childhood. Approximately 9.6–10.3% of all epilepsy cases in children are determined to be BRE and typically onsets between 2–13 years of age without any neurological or intellectual deficit before onset. Seizure types of BRE are characterized by brief hemitonic or hemiclonic movements localized in the orofacial region, often preceded by unilateral parasthesia, and anarthria or speech arrest and drooling of saliva due to sialorrhoea and saliva pooling [140]. Seizures occur mainly during sleep or upon awakening with or without a secondary generalization and remit during adolescence. Seizure frequency is usually low. EEG shows a high voltage spike (greater than 100 mV) focus located in the centro-temporal (Rolandic) area with normal background activity and epileptiform discharges actively occurring during sleep, which are diagnostic. Postictal weakness in the involved areas may occur [141].

## 8. Do stimulants Increase the Risk of Epilepsy?

### 8.1. Stimulants Are Highly Efficacious in CWE with Comorbid ADHD

A cumulative body of evidence indicates a high response rate of ADHD patients to stimulants, either alone or in combination, with well-tolerable safety profiles [142]. Therefore, the current guidelines recommend first-line methylphenidate (MPH) or amphetamine (AMP) as the stimulants of choice for children with ADHD [143]. The efficacy of stimulants was consistent in CWE and comorbid ADHD [144,145]. Actions of MPH contain inhibition of the dopamine and norepinephrine transporter, activation of the serotonin type 1A receptor, and redistribution of the vesicular monoamine transporter 2(VMAT2). Actions of AMP contain the block of the dopamine and noradrenaline transporter, VMAT2, and monoamine oxidase activity [146]. Approximately 10–30% of ADHD may not have a proper response to stimulants or may not be capable of tolerating side effects, including appetite suppression, mood difficulties, sleep disturbance, and tic exacerbation. Alternative stimulants or non-stimulants can be used in such cases [147].

### 8.2. MPH in Pediatric Subjects and AMP in Adults Are the First-Line ADHD Therapy

Two double-blind trials show that MPH achieved a response rate of 60–70% in CWE and comorbid ADHD [148,149] (Table 3); MPH was found to improve attention and memory, executive function, cognition, and the overall health-related quality of life [150]. The effect of MPH on cognition and memory was found to be dose-dependent [151]. The efficacy of stimulants was also evident in patients with refractory epilepsy. In two controlled trials, MPH had a response rate of 63–73% in children with severe epilepsy three months after treatment, regardless of the degree of learning disability [152,153]. Although AMP achieved a response rate of 48–58% in adults with ADHD [154,155], however, the evidence for AMP in CWE is less established, with a single-center retrospective study showing that the response to AMP was only 24% [156]. Cortese et al. compared the effectiveness and side effects of medications for the treatment of ADHD, including AMP and MPH through conducting a network meta-analysis and systematic review, and concluded that MPH in children and adolescents and AMP in adults are preferred first choice for short-term ADHD treatment [142].

### 8.3. Evaluating the Risk of Increasing Seizures in CWE and Comorbid ADHD Receiving Stimulants Is Crucial

Based on the findings of case reports [157,158,159] and a study pointing out possible seizure attacks in MPH overdose [160], stimulants were hypothesized to reduce the seizure threshold and, hence, increase the seizure frequency or even induce new seizures in patients with a long-standing seizure-free status [158,161]. An open-label trial studied 30 ADHD children with epilepsy receiving a single or two ASMs for an 8-week observation, and a single dose of MPH was given to all patients for the following 8 weeks. Although none of the 25 children with seizure-free status, however, there was a trend of increased seizure numbers in children with an active seizure at enrolment [162]. The same findings were reported by controlled trials and retrospective chart reviews [149,163]. The risk of seizure was also reported to be dose-dependent [156]. A population-based study retrieved the data of 29,604 ADHD children on MPH from the Hong Kong Clinical Data Analysis and Reporting System. Although the incidence of new-onset seizures in this report was only 0.2% (4.4 per 10,000 patient-years), seizure risk increased during the first month after treatment but not after three months when compared to pre-MPH [164]. Therefore, regulatory agencies indeed issued a caution for the use of stimulants in CWE and comorbid ADHD (i.e., www.fda.gov (https://www.accessdata.fda.gov/drugsatfda_docs/label/2013/010187s077lbl.pdf, accessed on 2 March 2023), “Prescriber’s Digital Reference”, www.pdr.net (https://www.pdr.net/drug-summary/Ritalin-LA-methylphenidate-hydrochloride-1003, accessed on 2 March 2023)). Despite the high efficacy of stimulants in ADHD, even in CWE, and in refractory epilepsy, many treating clinicians are reluctant to prescribe stimulants in CWE with comorbid ADHD due to the concern of the increased risk of seizure. Therefore, evaluating the risk of increasing seizure attacks in children with ADHD receiving stimulants, especially in CWE with refractory seizures, is crucial, because these patients have the highest risk of comorbid ADHD.

### 8.4. The False Myth of *Increased Risk of Seizures* with *Stimulants*

Subsequent evidence with larger samples and a more controlled design showed no increase in the risk of seizure in CWE and comorbid ADHD (Table 3). In a retrospective study, the comparisons between the seizure frequencies in the 3 months before, during, and after MPH treatment in 30 children and adults with active post-traumatic seizures showed a trend toward lesser seizure attacks during MPH treatment [165]. Additionally, several published literature consistently showed no increase in the risk of seizures in patients with or without poorly-controlled epilepsy [144,148,150,153,156,164,166,167,168,169,170]. In a previous report, only two children reported an increased seizure frequency out of the included 22 patients with poorly controlled epilepsy [153]. Liu et al. conducted a retrospective chart review on 18,166 stimulant users and 54,197 non-users in CWE. The results indicated no difference in the risk of seizure-related hospitalization between users and non-users [171]. As these results were consistent in patients with complex epilepsy, concerns regarding increased seizure frequency in children with complex or uncontrolled seizures and combined ADHD [149,162] can be relieved based on these data. In addition, a review of seven prospective studies concluded that MPH does not increase the seizure frequency or incidence of new-onset seizures [172]. Although the review was limited by the small number of pooled patients, short follow-up duration, and lack of a control group, studies with a larger sample size and longer duration follow-up were deemed necessary. While the evidence supports the safety of stimulants in CWE, it is still limited by the small sample size and uncontrolled design. Thus, safety data should be further studied in randomized, double-blinded, placebo-controlled trials.
ijms-24-05270-t003_Table 3Table 3Main Findings of Studies Reporting the Risk of Seizure in Patients on Stimulants.AuthorYearDesignPatientsStimulants/ADHD Improved the RateNo.Follow-UpMain FindingsLiu [171]2018RetrospectiveCWEMPH or AMP/NA72,3631 yearNo risk increaseWiggs [169]2017RetrospectiveADHD with and without epilepsyASC, d-MPH, DAS, LDX, MAH, MP, and MPHH/NA801,83810 yearsReduced riskRheims [150]2016ProspectiveActive or controlled CWE and comorbid ADHDMPH/75%16712–16 weeksNo risk increaseRadziuk [164]2015Open-label trialActive CWE and comorbid ADHD MPH/NA306 monthsNo risk increaseGonzalez- Heydrich [156]2014RetrospectiveControlled CWE and comorbid ADHDMPH/63% or AMP/24%3649 monthsNo risk increasedSantos [153]2013Open-label trialActive epilepsy and comorbid ADHDMPH/73%224 weeksNo risk increasedKoneski [144] 2011Open-label trialActive epilepsy and comorbid ADHDMPH/70.8%246 monthsNo risk increased Gonzalez- Heydrich [149] 2010Double-blind, controlled, trialActive epilepsy and comorbid ADHDOROS MPH/60–70%331–3 weeksIncreased riskYoo [170]2009Open-label trialControlled CWE and comorbid ADHDOROS MPH/64%258 weeksNo risk increase in 92% of patientsVan der Feltz-Cornelis [168]2006Open-label trialActive epilepsy and comorbid ADHDMPH/100%66 weeksNo risk increaseGucuyener [166]2003Open-label trialActive CWE and comorbid ADHDMPH/77%11912 monthsNo risk increaseHemmer [167]2001RetrospectiveADHDMPH, DAS, or ASC/NA2346 yearsNo risk increaseGross-Tsur [162]1997Open-label trialControlled CWE and comorbid ADHDMPH/70%308 weeksIncreased risk in active patientsWroblewski [165]1992Retrospectivechildren and adults with active post-traumatic seizuresMPH/NA303 monthsRisk reductionFeldman [148]1989Double-blind, controlled, trialControlled CWE and comorbid ADHDMPH/70%108 weeksNo risk increaseMPH: Methylphenidate; AMP: Amphetamine; CWE: Children with epilepsy; ADHD: Attention-deficit/hyperactivity disorder; NA: Not available. ASC: amphetamine salt combination; D-MPH: dexmethylphenidate hydrochloride; DAS: dextroamphetamine sulfate; LDX: lisdexamfetamine dimesylate; MAH: methamphetamine hydrochloride; MPH: methylphenidate; MPHH: methylphenidate hydrochloride.


## 9. ADHD and Epilepsy: Challenges and Opportunities

### 9.1. ADHD Is A Spectrum Disorder

ADHD is a neurobiological spectrum disorder with underlying multiple, complex, pathophysiological mechanisms. As previously mentioned, several genetic, environmental, and anatomic factors contribute to the liability of ADHD; children with ADHD also show smaller cerebellum, cortical, and other brain structures than healthy children, which is involved in reduced white matter tract connectivity. Emerging evidence also suggested the involvement of several neurotransmitters in the pathogenesis of ADHD, with an established dysregulation in the dopaminergic and adrenergic pathways [173]. In return, CWE and comorbid ADHD can show substantial variations in their phenotypes, and there is a need to view ADHD not as a unitary disorder but rather as a “dimensional marker that points to a spectrum of related disorders” [151]. This concept is supported by the growing evidence that ADHD presentations are qualitatively different to the extent that they should be considered distinct disorders due to the overlap in both executive and non-executive function deficits that can hardly distinguish ADHD traits [174,175]. The “dimensional” nature of ADHD can be challenging during the diagnosis and management; the difficulties in identifying the specific disorder of the ADHD spectrum can result in under or over-prescribing the pharmacological options. The challenge is expected to be even higher in patients with other disorders, such as CWE [151].

### 9.2. Challenges in the Early Diagnosis of ADHD in CWE

Early identification and interventions are crucial to optimize the developmental outcomes of children with ADHD. Previous reports showed that early treatment is associated with better ADHD-related symptoms, developmental outcomes, and quality of life. Additionally, early treatment demonstrated cost-effectiveness and reduced direct and indirect healthcare resource utilization [176]. Despite ADHD being a common comorbidity in CWE, however, there is lacking a guideline regarding screening, specific diagnostic methods, and proper treatment. The current guidelines state that a multidisciplinary team should diagnose ADHD after excluding other causes; the ADHD symptoms should be assessed through a validated rating scale filled by the parents and teachers [177,178]. Several rating scales are currently available that evaluate various aspects of the child’s activities; nonetheless, many of them still lack standardization in clinical practice. In the 2017 ILAE consensus, the authors concluded that limited evidence is available regarding the diagnostic yield of the ADHD rating scale in the setting of CWE. The currently available scales suffer from low sensitivity in identifying inattentive presentation of ADHD—a common presentation in CWE- and low specificity, which donates that these tools can be used for screening purposes only and should be complemented by a detailed psychiatric assessment [91]. The scales also exhibited limited utility and were not validated in some racial groups and patients with intellectual disabilities [179,180]. To provide a practical and evidence-based guide in the managing of CWE, a working team under the ILAE Pediatric Commission targeting screening, diagnosis, and management of ADHD in CWE through a multi-trait, computerized, multi-method approach yielded a high positive predictive value in children with ADHD. Still, these approaches have neither been validated in larger multicenter studies nor in CWE [91].

The diagnosis of ADHD can be challenged in some seizure types, such as CAE. Up to two-thirds of CAE patients can have comorbid ADHD [139]. Patients with controlled CAE can develop cognitive defects and variable levels of inattentiveness secondarily to impaired consciousness, resembling ADHD; patients with CAE were found to have difficulties in visual attention, visuospatial skills, learning, memory, and language [181]. ASMs can further burden visual attention, psychomotor, processing, and other cognitive functions in children with CAE [182]. Therefore, ADHD may be missed in children with CAE. The ILAE consensus provided a set of recommendations to differentiate between ADHD and CAE based on symptoms and EEG with hyperventilation [91]. Learning disabilities and sleep disorders are common in CWE, which can present with inattentiveness. Thus, ADHD may be misdiagnosed in these populations [183,184].

### 9.3. Treatment Strategy for ADHD in Epilepsy

The treatment strategy for ADHD in epilepsy involves pharmacological management and behavioral therapy.

#### 9.3.1. Pharmacological management

##### Choosing Proper ASMs

There are several ASMs showing effective both in epilepsy and ADHD. For example, carbamazepine or oxcarbazepine is reported to improve ADHD-related behavioral and mood problems [129,185,186] and inattention [187] in patients with partial epilepsy; levetiraacetm is reported to successfully treat ADHD with nocturnal focal epileptic discharges [188]; topiramate is reported to positively improve behavioral changes in pediatric epilepsy with ADHD [163]. Although those medications may be beneficial to CWE with ADHD, clinicians should perform periodic reassessments to adjust the dosages of ASMs and/or ADHD medications according to patients’ condition or to titrate the ASMs polytherapy or replace a new ASM with less psychological and cognitive effect.

##### Pharmacological Management for ADHD in Epilepsy

Pharmacological management includes stimulants and non-stimulants.

##### Stimulants

Stimulants, including MPH and amphetamines, are more efficacious, and act faster, whereas non-stimulants are less efficacious and take much time for their onset of action. The action mechanisms for MPH are through reuptake inhibition of dopamine, norepinephrine, and amphetamines, which act primarily through modulating the release of dopamine and norepinephrine [189,190]. However, only a few case studies reported new-onset seizures in the treatment of ADHD (Table 3). In fact, occurrences of seizures were not significantly different between stimulants and placebo [191]. MPH is the optical choice for treating CWE with comorbid ADHD [192]. If patients with epilepsy are poorly controlled, and stimulants cannot control comorbid ADHD with proper doses, non-stimulants should be considered.

##### Non-Stimulants

Non-stimulants currently approved for ADHD by the FDA are broadly classified as (1) monoamine reuptake (transporter) inhibitors (Atomoxetine); (2) receptor modulators (guanfacine, clonidine); and (3) multimodal agents.

Atomoxetine is a selective noradrenergic reuptake inhibitor with a high affinity for presynaptic norepinephrine transporters (NETs) [193,194]. Atomoxetine was approved for the treatment of ADHD based on a series of double- blind, randomized controlled trials in children ≥6 years of age, adolescents, and adults [195,196,197,198]. Adverse effects include, abdominal pain, anorexia, nausea, decreased appetite, diarrhea, severe liver injury, vomiting, dizziness, drowsiness, fatigue, headache, insomnia, sedation, suicidal ideation, mild increases in blood pressure and heart rate, decreased libido, and dysuria [142,193]. If MPH and atomoxetine fail, guanfacine or clonidine may deserve a trial [199]. Guanfacine, which is not a stimulant, is a selective alpha-2A agonist that is currently FDA-approved as a supplementary therapy to stimulants for the treatment of ADHD in children and adolescents ages 6–17 [200]. Guanfacine is orally administered once daily, usually in the morning, but it should not be taken with high-fat meals. The action mechanism for Guanfacine in ADHD is unknown, but it may work on certain receptors in the prefrontal cortex, a part of the brain where behaviors related to ADHD, such as inattention and impulsiveness, are thought to be controlled [201]. Clonidine, an α _2_ –agonist, is approved for hypertension in adults and is relatively safe and well-tolerated [202]. It can modulate sympathetic tone by increasing noradrenergic outflow from the locus coeruleus to the prefrontal cortex and by directly stimulating of presynaptic alpha-2A receptors in the cortex, leading to increased attention regulation and behavior, suggesting a positive therapeutic effect on ADHD [203]. Although several studies support the role of clonidine in the potential treatment of children with ADHD [204,205,206], this medication is not as effective as MPH in improving ADHD symptoms [199]. Clonidine is often prescribed and dosed after dinner or at bedtime for its sedating effects. Adverse effects include somnolence, fatigue, bradycardia, and hypotension [203].

##### Non-Pharmacological Management: Behavioral Therapies

Few research assessed behavioral therapies for ADHD in children with comorbid epilepsy. Although behavioral therapies may not show substantial differences in therapeutic efficacy in children with ADHD with and without epilepsy, however, in CWE, compared to otherwise healthy children with ADHD, there may be a higher likelihood for medical contraindications or parental expectation to decrease the number of medications [207]. It should be noted that behavior therapy with parent training is the only non-pharmacological method that showed statistically significant results in clinical trials, and a combination of behavior therapy with stimulants was more effective than stimulants alone [208], suggesting that parental participation is a key factor in determining behavior therapies will be successful or not.

#### 9.3.2. Drug Interactions

Atomoxetine has been shown to have no interactions with any AEDs [209], while MPH can increase serum phenytoin concentrations [210]. Additionally, concurrent administration of carbamazepine may reduce MPH serum concentrations, resulting in reduced efficacy [192].

### 9.4. Novel ASMs against Both Epilepsy and ADHD

Lamotrigine (6-(2,3-dichlorophenyl)-1,2,4-triazine-3,5-diamine; molecular formula: C9H7N5Cl2; molecular weight: 256.09), developed by the GlaxoSmithKline, is a new generation broad-spectrum ASM for epilepsy and bipolar disorder. Lamotrigine was first approved by the US Food and Drug Administration (FDA) in 1994. Over the years, lamotrigine has been approved for the treatment of epilepsy and bipolar disorder [211]. Guidelines have recommended this medication as first-line monotherapy or polytherapy for focal and generalized seizures in children and adults [212,213,214,215,216]. Mechanisms of the antiepileptic effect of lamotrigine include blocking voltage-gated sodium channels and voltage-gated calcium channels, regulating the release of the inhibitory and excitatory neurotransmitters, stabilizing membrane potential of neurons, abolishing the repetitive firing of neurons [217,218,219].

Several studies show that lamotrigine can improve epilepsy without worsening ADHD symptoms [220,221,222,223]. It is not clear how lamotrigine can exert favorable effects on ADHD symptoms in CWE. The anti-epileptic effect of this medication may mainly contribute to this positive effect in improving ADHD in CWE.

### 9.5. ASMs in CWE and Comorbid ADHD

Comorbid ADHD is usually present in children with complex epilepsy who tend to be on long-term treatment. Few studies have examined the impact of ASMs on behavioral and cognitive outcomes in CWE and comorbid ADHD. Phenobarbital, valproic acid, and ethosuximide increased ADHD symptoms, particularly attention deficit. On the other hand, published reports showed inconsistent results regarding the impact of other ASMs on cognitive functions and ADHD symptoms, with some small studies showing a beneficial role of lacosamide and carbamazepine on behavior [104]. It is still also unclear how polypharmacy, with multiple pharmacokinetic interactions, can affect the behavioral outcomes in CWE and comorbid ADHD. In the ILAE consensus, it was concluded that valproate and polypharmacy are associated with worsening inattentive symptoms in CWE. However, these findings should be cautiously interpreted as polytherapy may indicate complicated epilepsy rather than a cause of deteriorating ADHD symptoms [91].

### 9.6. Recent Medications Development for ADHD

Stimulants are the first-line treatment for ADHD. However, numerous clinicians, care providers, patients’ families, and teachers notice that some patients may have a good initial response to stimulants, but the effects are lessened [224]. Due to the development of tolerance, variability in patient response to specific pharmaceutics agents, and unwanted side effects related to stimulants, the need for relief of ADHD drives research for new treatments targeting the core symptoms of ADHD and fewer side effects. New medications such as lisdexamfetamine, viloxazine, serdexmethylphenidate/d-MPH, and mazindol have been approved for ADHD. Centanafadine, dasotraline, vortioxetine, droxidopa, and baicalin have not been approved by FDA and are under investigation.

#### 9.6.1. Lisdexamfetamine (LDX)

LDX is a long-acting amfetamine prodrug [225]. Catabolic products of LDX are d-amphetamine and L-lysine. LDX is safe and well-tolerated. LDX is approved for children, adolescences, and adults with ADHD because the efficacy of LDX in ADHD has been successfully investigated in several large-scale, double-blind, randomized clinical trials [226,227,228,229,230,231]. Adverse effects include decreased appetite, headache, insomnia, irritability, and weight reduction [232].

#### 9.6.2. Viloxazine

Viloxazine is a selective norepinephrine reuptake inhibitor with activity in the noradrenergic and serotonergic pathways [233], is approved to be used in the treatment of ADHD children and adolescents and is administered orally in once-daily oral doses [234]. One possible action mechanism for viloxazine against ADHD is through increased efflux of norepinephrine and dopamine in the prefrontal cortex [235]. Viloxazine has no drug interactions with LDX or MPH [233,236]. Adverse effects include somnolence, sedation, headache, fatigue, decreased appetite, abdominal pain, upper respiratory infection, nausea and vomiting, cardiovascular effects and suicidal ideation.

#### 9.6.3. Serdexmethylphenidate (SDX)

Serdexmethylphenidate (SDX) structurally consists of d-methylphenidate connected to a nicotinoyl-L-serine molecule via a carboxymethylene linker. SDX has no affinity for DATs, NETs or serotonin transporters (SERTs). Although SDX is a novel prodrug of d-methylphenidate (d-MPH) and is pharmacologically inactive until gradually converted to active d-MPH in the lower intestinal tract, FDA has not approved it as a single entity for any indication [237]. As the immediate-release d-MPH component accounts for the rapid increase in plasma MPH concentrations, while the SDX component increases the MPH concentration through the evening hours, a multicenter, randomized, double-blind, placebo-controlled laboratory classroom study, SDX/d-MPH was shown to be efficacious and well tolerated in children aged 6–12 years with ADHD [238]. Therefore, SDX/d-MPH has been approved by FDA. Adverse effects include decreased appetite, nausea, vomiting, dyspepsia, abdominal pain, decreased weight, anxiety, dizziness, irritability, tachycardia, and increased blood pressure.

#### 9.6.4. Mazindol

Mazindol, originally designed as an appetite suppressant, displays unique pharmacologic activities that include NET, DAT, and SERT inhibition [239], and modulation of serotonin (5-HT_1A_, 5-HT_7_), muscarinic, histamine H_1_, μ-opioid, and orexin-2 receptors [240]. Since mazindol’s potency as a NET inhibitor is similar to atomoxetine, mazindol is presumed to be effective in the treatment of ADHD. A study concluded that the effect of mazindol was similar to d-amphetamine and nicotine by PET imaging in human subjects [241]. Mazindol has been shown to be effective in ADHD with good safety in children [242] and adults [240]. Adverse effects include fatigue, decreased appetite, weight loss, drowsiness, dry mouth, nausea, constipation, intestinal distension and upper abdominal pain, and increases in blood pressure and heart rate.

#### 9.6.5. Centanafadine

Centanafadine (EB-1020), a norepinephrine and dopamine reuptake inhibitor, is not a stimulant like MPH. Centanafadine can increase striatal dopaminergic neurotransmission, differentiating it from the noradrenergic ADHD drugs, atomoxetine and viloxazine [194]. Since this medication was found to increase dopamine and norepinephrine in the prefrontal cortex and striatum of animals [194], and it was also effective in preventing hyperactivity in the neonatal 6-hydroxydopamine brain lesion model of ADHD [194], centanafadine was therefore presumed to be a potential medication for the treatment of ADHD. Two Phase II clinical trials of centanafadine treatment significantly improve adults with ADHD [243]. Adverse effects include decreased appetite, nausea, insomnia, fatigue and dry mouth.

#### 9.6.6. Dasotraline

Dasotraline [(1R, 4S)-4-(3, 4-Dichlorophenyl)-1, 2, 3, 4-tetrahydronaphthalen-1-amine] is a dual dopamine and norepinephrine reuptake inhibitor (DNRI). However, human PET studies demonstrated its preferential inhibition of dopamine transporters (DAT) and NET and weaker inhibition of serotonin transporters [244,245]. Several studies have shown a good effect of dasotraline in improving ADHD symptoms and the safety of this medication in the general population [246,247,248]. Adverse effects include insomnia, irritability, appetite suppression, decreased weight, dry mouth, anxiety, panic attack, hallucinations, and delusions were reported [246].

#### 9.6.7. Vortioxetine (VTX)

VTX (1-[2-(2,4-dimethylphenylsulfanyl)phenyl]piperazine) is approved as an antidepressive medication with multiple mechanisms of action: serotonin reuptake inhibitor, 5-HT7, 5-HT3 and 5-HT1D receptor antagonist, 5-HT1B receptor partial agonist, and 5-HT1A receptor agonist, leading to modulation of histamine, norepinephrine, acetylcholine, glutamate, and γ-aminobutyric acid (GABA) [249,250]. An animal study shows that VTX significantly increased the extracellular concentration of 5-HT in the prefrontal cortex with mild increases in norepinephrine and dopamine [250]. VTX can reverse phencyclidine-induced deficits in attentional set-shifting [251]. There are two cases with ADHD comorbid learning difficulties in major academic domains, and wider symptoms of sickness behavior reported to be responsive to MPH in combination with Vortioxetine [252]. However, a phase II random control trial did not detect a better effect of VTX than placebo in improving ADHD symptoms [253]

#### 9.6.8. Droxidopa

Droxidopa (L-*threo*-dihydroxyphenylserine), is chemically analogous to levodopa. Droxidopa is a CNS-penetrant norepinephrine prodrug that is metabolized by DOPA decarboxylase to increase extracellular concentrations of norepinephrine in the brain [254]. The efficacy of droxidopa is predicted to be similar to the α_2A_-adrenoceptor agonists. Droxidopa is approved for adults with neurogenic orthostatic hypotension associated with primary autonomic failure (such as Parkinson’s disease, multiple system atrophy or pure autonomic failure), dopamine β-hydroxylase deficiency or nondiabetic autonomic neuropathy) and is administered orally [255]. Droxidopa is reported to significantly decrease the ADHD-RS-Total score [256]. Adverse events include abnormal dreams, depressed mood, headache, insomnia, somnolence, sedation, suicidal ideation, musculoskeletal stiffness and myalgia, hyperhidrosis, nausea, and cough [256].

#### 9.6.9. Baicalin

Scutellaria baicalensis Georgi (SBG), a Chinese traditional medicine is used to treat fever, hepatitis, hypertension, inflammation, and jaundice [257]. Baicalin (or baicalein), a flavonoid purified from SBG, can protect against several neurotoxins, which employ DAT [258,259]. Baicalin is found to decrease hyperactivity in the spontaneously hypertensive rat model and increase markers of striatal dopamine function. Therefore, it is proposed to be a potential medication for the treatment of ADHD [260].

### 9.7. Impact of Comorbid ADHD on Developmental Outcomes

Previous reports showed that CWE, particularly complex types, suffer from impaired sustained attention, working memory, processing speed, and neuroanatomical derangements [78]. Imaging studies demonstrated that children with focal epilepsy had significantly thinner cortical thickness beyond the focal seizure area and slower white matter expansion; both abnormalities were significantly correlated with lower performance and verbal intelligence quotient (IQ), alongside lower executive functions than normal control [261,262]. On the other hand, ADHD patients showed relative cortical thinning compared to normal control, which correlated with worse outcomes in the global assessment scales and remission [263]. Thus, it can be hypothesized that CWE and comorbid ADHD may exhibit neuroanatomical abnormalities, leading to worse cognitive and intellectual outcomes.

### 9.8. Potential Role of Precision Medicine

Precision medicine is an emerging field that aims to tailor the treatment of disorders with genetic backgrounds according to the detected genetic mutation and functional alteration. The concept of precision medicine stems from the fact that genetic mutations affect the eligibility and response to treatment, as well as the safety outcomes [264]. In the setting of epilepsy, some genetic epilepsies were found to be associated with an increased risk of recurrence and sudden death [265]. As the growing body of evidence shows a shared genetic background between epilepsy and ADHD, precision medicine can play a role in tailoring more effective treatments that optimize long-term developmental and behavioral outcomes.

## 10. Conclusions

The association between CWE and comorbid ADHD is still under-recognized in clinical practice, even though the well-established bidirectional connection and shared genetic/non-genetic factors between epilepsy and comorbid ADHD largely rule out the possibility of a chance in this association. Stimulants are effective in children with comorbid ADHD, and the current body of evidence supports their safety in CWE within the approved dose. Early identification and proper management of comorbid ADHD are crucial to optimize the prognosis and reduce the risk of adverse long-term neurodevelopmental outcomes.

## Figures and Tables

**Figure 1 ijms-24-05270-f001:**
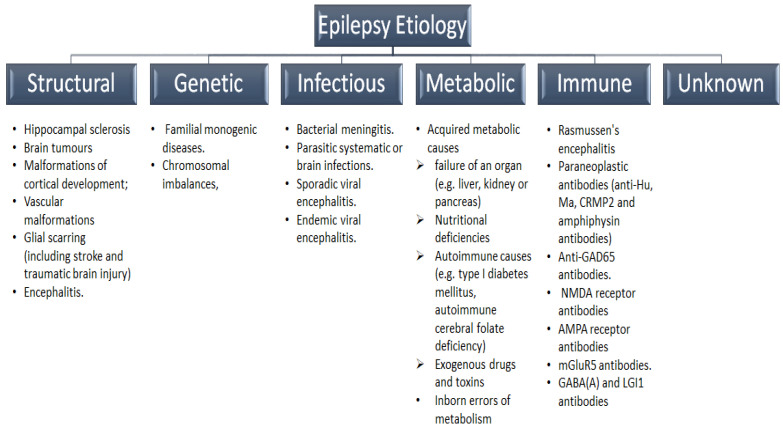
Etiology of epilepsies.

**Table 1 ijms-24-05270-t001:** Key symptoms of ADHD.

Inattentive Symptoms	Hyperactivity/Impulsivity Symptoms
Pays no attention to details and makes reckless actions	❖ Excessive fidgets.
Inability to maintain attention in school, exercise, and work	Failure to remain in the seat when required.
When addressed, he does not appear to listen.	Frequently runs around or climbs in inappropriate settings.
Inability to follow directions, failure to complete work	Often unable to quietly participate in play when needed.
Demonstrate a significant level of disorganization during the execution of tasks and activities	Always “on the move” and appears “powered by a motor.”
Tendency to avoid tasks that require attention, such as reading long papers.	Talks excessively.
Items required for tasks and activities are misplaced.	Answers impatiently before the inquiry is finished.
Extraneous stimuli such as irrelevant ideas cause distraction.	Frequently unable to await their turn.
Inattention to regular tasks such as paying bills and making appointments	Frequently interrupts or interferes with others

**Table 2 ijms-24-05270-t002:** Prevalence of ADHD in CWE.

Country	Published Year	ADHD/CWE%	References
Brazil	2010	29.1	[82]
China	2012, 2018	24.7–42	[81,84]
Israel	2011	70	[79]
Iran	2016	60.4	[83]
Norway	2011, 2016	12.1–31.7	[42,49]
UK	2014, 2003	12–33	[78,80]
USA	2012	23	[52]
USA	2003	37.3	[13]

## Data Availability

Not applicable because this review did not report any new data.

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
