# Peer review of "Epilepsy and Attention Deficit Hyperactivity Disorder: Connection, Chance, and Challenges"

_ijms, 2023, doi:10.3390/ijms24065270_

Round 1

Reviewer 1 Report

Fan et al. reviewed the relationship between epilepsy and attention deficit hyperactivity disorder. The manuscript is interesting and well-written.

I have some minor concerns which need to be addressed before the publication

1.     The authors should include a brief description about the treatment strategy for ADHD in epilepsy

2.     The authors are advised to discuss ADHD in different types of epilepsy using clinical data.

3.     Recent drug development for ADHD should also be discussed

4.     Some anti-epileptic drugs have shown efficacy against ADHD, such as Lamotrigine. Hence, the Authors should incorporate such information, including the mechanism of action that may improve the novelty of this ms.  

5.     The conclusion part should be revised; it is too long.

Author Response

Reviewer A

Fan et al. reviewed the relationship between epilepsy and attention deficit hyperactivity disorder. The manuscript is interesting and well-written.

I have some minor concerns which need to be addressed before the publication

Q1: The authors should include a brief description about the treatment strategy for ADHD in epilepsy

R: Thank you. We have added your suggestions in our new MS.

9.3. Treatment strategy for ADHD in epilepsy

The treatment strategy for ADHD in epilepsy involves pharmacological management and behavioral therapy.

9.3.1.Pharmacological management

9.3.1.1. Choosing proper ASMs

There are several ASMs showing effective both in the epilepsy and ADHD. For example, carbamazepine or oxcarbazepine is reported to improve ADHD-related behavioral and mood problems[129,185,186] and inattention[187] in patients with partial epilepsy; levetiraacetm is reported to successfully treat ADHD with nocturnal focal epileptic discharges[188]; topiramate is reported to positively improve behavioral changes in pediatric epilepsy with ADHD[163]. Although those medications may bebeneficial to CWE with ADHD, clinicians should perform periodic reassessments to adjust the dosages of ASMs and/or ADHD medications according to patients’ condition or to titrate the ASMs polytherapy or replace a new ASM with less psychological and cognitive effect.

9.3.1.2. Pharmacological management for ADHD in epilepsy

Pharmacological management includes stimulants and non-stimulants.

9.3.1.2.1. Stimulants

Stimulants, including MPH and amphetamines, are more efficacious, and act faster, whereas non-stimulants are less efficacious and take much time for their onset of action. The action mechanisms for MPH are through reuptake inhibition of dopamine, norepinephrine, and amphetamines, which act primarily through modulating the release of dopamine and norepinephrine [189,190]. However, only a few case studies reported new-onset seizures in the treatment of ADHD (Table 3). In fact, occurrences of seizures were not significantly different between stimulants and placebo[191]. MPH is the optical choice for treating CWE with comorbid ADHD[192]. If patients with epilepsy are poorly controlled, and stimulants cannot control comorbid ADHD with proper doses, non-stimulants should be considered.

9.3.1.2.2. Non-stimulants

Non-stimulants currently approved for ADHD by the FDA are broadly classified as (1) monoamine reuptake (transporter) inhibitors (Atomoxetine); (2) receptor modulators (guanfacine, clonidine); and (3) multimodal agents.

Atomoxetine is a selective noradrenergic reuptake inhibitor with a high affinity for presynaptic norepineph- rine transporters (NET)[193,194]. Atomoxetine was approved for the treatment of ADHD based on a series of double- blind, randomized controlled trials in children ≥6 years of age, adolescents, and adults[195-198]. Adverse effects include vomiting, abdominal pain, nausea, decreased appetite, diarrhea, dizziness, headache, insomnia, sedation, drowsiness, fatigue, anorexia, suicidal ideation, severe liver injury, mild increases in blood pressure and heart rate, decreased libido, and dysuria[142,193]. If MPH and atomoxetine fail, guanfacine or clonidine may deserve a trial[199]. Guanfacine, which is not a stimulant, is a selective alpha-2A agonist that is currently FDA-approved as a supplementary therapy to stimulants for the treatment of ADHD in children and adolescents ages 6–17[200]. It is orally administered once daily, usually in the morning, but it should not be taken with high-fat meals. The mechanism of action in ADHD is unknown, but it may work on certain receptors in the prefrontal cortex, a part of the brain where behaviors related to ADHD, such as inattention and impulsiveness, are thought to be controlled[201]. Clonidine, an α 2 –agonist, is approved for hypertension in adults and is relatively safe and well-tolerated [202]. It can modulate sympathetic tone by increasing noradrenergic outflow from the locus ceruleus to the prefrontal cortex and by directly stimulating presynaptic alpha-2A receptors in the cortex, leading to increased attention regulation and behavior, suggesting a positive therapeutic effect on ADHD[203]. Although several studies support the role of clonidine in the potential treatment of children with ADHD[204-206], this medication is not as effective as MPH in improving ADHD symptoms[199]. Clonidine is often prescribed and dosed after dinner or at bedtime for its sedating effects. Adverse effects include somnolence, fatigue, bradycardia, and hypotension[203].

9.3.1.3. Non-pharmacological management: Behavioral therapies

Few research assessed behavioral therapies for ADHD in children with comorbid epilepsy. Although behavioral therapies may not show substantial differences in therapeutic efficacy in children with ADHD with and without epilepsy, however, in CWE, compared to otherwise healthy children with ADHD, there may be a higher likelihood for medical contraindications or parental expectation to decrease the number of medications[207]. It should be noted that behavior therapy with parent training is the only non-pharmacological method that showed statistically significant results in clinical trials, and a combination of behavior therapy with stimulants was more effective than stimulants alone[208], suggesting that parental participation is a key factor in determining behavior therapies will be successful or not.

9.3.2. Drug interactions

Atomoxetine has been shown to have no interactions with any AEDs[209], while MPH can increase serum phenytoin concentrations[210]. Additionally, concurrent administration of carbamazepine may reduce MPH serum concentrations, resulting in reduced efficacy[192].

Q2. The authors are advised to discuss ADHD in different types of epilepsy using clinical data.

R: Thank you. We have added new sentences in our revised MS.

  1. 7. ADHD in different types of epilepsy

The prevalence of ADHD is significantly higher in CWE than in normal controls (Table 2). Because it is not clear whether the cause for the connection between ADHD and CWE is due to a global attention deficit or a specific network deficit, therefore, the connection between ADHD and CWE has been presumed to be attributed to interactions between ongoing seizures, underlying causes of symptomatic epilepsy or innate properties of epilepsy[122-125]. There are several types of epilepsy associated with ADHD discussed in the following.

7.1. Generalized epilepsy

Generalized seizures start when all brain areas are affected by an abnormal, widespread, excessive electrical impulse. Symptoms include lossing muscle tone, clonic, jerking, or stiffening limbs, staring and blinking, and rhythmic, full-body jerking, typically provoked by sleep deprivation, fatigue, and alcohol drinking[126]. Antecedent and birth history, neurological examination, intelligence, and head size are typically normal. Most children with seizures were found to be in a generalized manner[127]. Attentional difficulties are more common among patients with generalized epilepsies than those with partial seizures[85,128]. Some of them also show memory impairment, learning problems, and psychomotor impairments[129]. The neurological basis for such cognitive impairments in patients with generalized epilepsy is unclear. Impaired brain development in children with epilepsy has been linked to a range of cognitive and behavioral issues, including the development of ADHD.

7.2. Frontal lobe epilepsy (FLE)

The frontal lobes, which constitute almost one-third of the human brain, are the largest parts of the brain. Frontal lobes can work with other brain regions and contribute to overall brain function. FLE is the second most common partial epilepsies in childhood[130]. Any epileptic discharges within the frontal regions may potentially affect frontal lobe-associated functions, such as motor functions, control of continence, and olfaction, voluntary eye movements, speech and language abilities, executive functions, motivational behaviors, and social competency. Impairments of those functions may lead to variable symptoms in patients with FLE [131]. The manifestations of FLE include nausea, vomiting, vertigo, motor and cognitive abnormalities, abnormal body posturing, tics, disinhibition, excessive excitement/irritability, inattention, and impulsivity[132]. The psychiatric manifestations of FLE may be difficult to differentiate from those with ADHD without epilepsy, and ADHD may mimic FLE with or without ADHD and vice versa [133]. As the underlying mechanisms for FLE are that frontal cortex abnormalities dysregulate the frontal lobe networks, leading to attention and hyperactivity at a crucial early stage of brain maturation[134], it may be easily confused with ADHD[132]. Nocturnal FLE is defined by seizures that attack in sleep and originate in the orbitofrontal or mesial frontal areas and has been reported to be associated with ADHD[135]. Patients with tuberous sclerosis involving frontal lobes were reported to present seizures with concomitant ADHD[136].

7.3. Childhood absence epilepsy (CAE)

CAE is one of the common forms of pediatric epilepsy. Approximately 10-17% of all childhood-onset seizures are determined to be typical CAE[137], which presents with a short duration (4 to 30 seconds), transient disturbance of consciousness, blinking, staring, and/or subtle or motionless activity without post-ictal symptoms, and less commonly, various types of automatisms, occurring numerous times per day and happen without notice for long. CAE is typically provoked by hyperventilation and sleep deprivation. EEG shows generalized 2.5 to 5 Herz spike-wave activity[138].  A study reports that 25% of CAE children have subtle cognitive deficits, and ADHD is significantly associated with CAE (26% of children with CAE vs 6% of normal children)[139], suggesting cognitive or psychiatric comorbidities, especially ADHD, are associated with CAE.

7.4. Benign rolandic epilepsy (BRE)

BRE, or benign epilepsy with centrotemporal spikes (BECTS), is a common idiopathic focal epilepsy in childhood. Approximately 9.6-10.3% of all epilepsy cases in children are determined to be BRE and typically onsets between 2-13 years of age without any neurological or intellectual deficit before onset. Seizure types of BRE are characterized by brief hemitonic or hemiclonic movements localized in the orofacial region, often preceded by unilateral parasthesia, and anarthria or speech arrest and drooling of saliva due to sialorrhoea and saliva pooling[140]. Seizures occur mainly during sleep or upon awakening with or without a secondary generalization and remit during adolescence. Seizure frequency is usually low. EEG shows a high voltage spike (greater than 100 mV) focus located in the centro temporal (Rolandic) area with normal background activity and epileptiform discharges actively occurring during sleep, which are diagnostic. Postictal weakness in the involved areas may occur[141].

Q3. Recent drug development for ADHD should also be discussed

R: Thank you for your excellent suggestion. We have added update information in our new MS.

9.6. Recent medications development for ADHD

Stimulants are the first-line treatment for ADHD. However, numerous clinicians, care providers, patient’s families, and teachers notice that some patients may have a good initial response to stimulants, but the effects are lessened[224]. Due to the development of tolerance, variability in patient response to specific pharmaceutics agents, and unwanted side effects related to stimulants, the need for relief of ADHD drives research for new treatments targeting the core symptoms of ADHD and fewer side effects. New medications such as lisdexamfetamine, viloxazine, serdexmethylphenidate/d-MPH, and mazindol have been approved for ADHD. Centanafadine, dasotraline, vortioxetine, droxidopa, and baicalin have not been approved by FDA and are under investigation.

9.6.1.Lisdexamfetamine (LDX)

LDX is a long-acting amfetamine prodrug[225]. Catabolic products of LDX are d-amphetamine and L-lysine. LDX is safe and well-tolerated. LDX is approved for children, adolescences, and adults with ADHD because the efficacy of LDX in ADHD has been successfully investigated in several large-scale, double-blind, randomized clinical trials [226-231]. Adverse effects include decreased appetite, headache, insomnia, irritability, and weight reduction[232].

9.6.2.Viloxazine

Viloxazine is a selective norepinephrine reuptake inhibitor with activity in the noradrenergic and serotonergic pathways[233], is approved to be used in the treatment of ADHD children and adolescents and is administered orally in once-daily oral doses[234]. The action of mechanisms for viloxazine against ADHD is probably through increased efflux of norepinephrine and dopamine in the PFC[235]. Viloxazine has no drug interactions with LDX or MPH[233,236]. Adverse effects include somnolence, sedation, headache, fatigue, decreased appetite, abdominal pain, upper respiratory infection, nausea and vomiting, cardiovascular effects and suicidal ideation.

9.6.3. Serdexmethylphenidate (SDX)

Serdexmethylphenidate (SDX) structurally consists of d-methylphenidate connected to a nicotinoyl-L-serine molecule via a carboxymethylene linker. SDX has no affinity for DAT, NET or SERT. Although SDX is a novel prodrug of d-methylphenidate (d-MPH) and is pharmacologically inactive until gradually converted to active d-MPH in the lower intestinal tract, FDA has not approved it as a single entity for any indication[237]. As the immediate-release d-MPH component accounts for the rapid increase in plasma MPH concentrations, while the SDX component increases the MPH concentration through the evening hours, a multicenter, randomized, double-blind, placebo-controlled laboratory classroom study, SDX/d-MPH was shown to be efficacious and well tolerated in children aged 6–12 years with ADHD[238]. Therefore, SDX/d-MPH has been approved by FDA. Adverse effects include decreased appetite, nausea, vomiting, dyspepsia, abdominal pain, decreased weight, anxiety, dizziness, irritability, tachycardia, and increased blood pressure.

9.6.4.Mazindol

Mazindol, originally designed as an appetite suppressant, displays unique pharmacologic activities that include NET, DAT, and SERT inhibition[239], and modulation of serotonin (5-HT1A, 5-HT7), muscarinic, histamine H1, μ-opioid, and orexin-2 receptors[240]. Since mazindol’s potency as a NET inhibitor is similar to atomoxetine, mazindol is presumed to be effective in the treatment of ADHD. A study concluded that the effect of mazindol was similar to d-amphetamine and nicotine by PET imaging in human subjects[241]. Mazindol has been shown to be effective in ADHD with good safety in children[242] and adults[240]. Adverse effects include fatigue, decreased appetite, weight loss, drowsiness, dry mouth, nausea, constipation, intestinal distension and upper abdominal pain, and increases in blood pressure and heart rate.

9.6.5. Centanafadine

Centanafadine (EB-1020), a norepinephrine and dopamine reuptake inhibitor, is not a stimulant like MPH. Centanafadine can increase striatal dopaminergic neurotransmission, differentiating it from the noradrenergic ADHD drugs, atomoxetine and viloxazine[194]. Since this medication was found to increase dopamine and norepinephrine in the PFC and striatum of animals [194], and it was also effective in preventing hyperactivity in the neonatal 6-hydroxydopamine brain lesion model of ADHD[194], centanafadine was therefore presumed to be a potential medication for the treatment of ADHD. Two Phase II clinical trials of centanafadine treatment significantly improve adults with ADHD[243]. Adverse effects include decreased appetite, nausea, insomnia, fatigue and dry mouth.

9.6.6.Dasotraline

Dasotraline [(1R, 4S)-4-(3, 4-Dichlorophenyl)-1, 2, 3, 4-tetrahydronaphthalen-1-amine] is a dual dopamine and norepinephrine reuptake inhibitor (DNRI). However, human PET studies demonstrated its preferential inhibition of dopamine transporters (DAT) and NET and weaker inhibition of serotonin transporters[244,245]. Several studies have shown a good effect of dasotraline in improving ADHD symptoms and the safety of this medication in the general population[246-248]. Adverse effects include insomnia, irritability, appetite suppression, decreased weight, dry mouth, anxiety, panic attack, hallucinations, and delusions were reported[246].

9.6.7. Vortioxetine (VTX)

VTX  (1-[2-(2,4-dimethylphenylsulfanyl)phenyl]piperazine) is approved as an antidepressive medication with multiple mechanisms of action: serotonin reuptake inhibitor, 5-HT7, 5-HT3 and 5-HT1D receptor antagonist, 5-HT1B receptor partial agonist, and 5-HT1A receptor agonist,  leading to modulation of histamine, norepinephrine, acetylcholine, glutamate, and γ-aminobutyric acid (GABA)[249,250]. An animal study shows that VTX significantly increased the extracellular concentration of 5-HT in the PFC with mild increases in norepinephrine and dopamine[250]. VTX can reverse phencyclidine-induced deficits in attentional set-shifting [251]. There are two cases with ADHD comorbid learning difficulties in major academic domains and wider symptoms of sickness behavior reported to be responsive to MPH in combination with Vortioxetine[252]. However, a phase II random control trial did not detect a better effect of VTX than placebo in improving ADHD symptoms[253]

9.6.8.Droxidopa

Droxidopa ( L-threo-dihydroxyphenylserine), is chemically analogous to levodopa. Droxidopa is a CNS-penetrant norepinephrine prodrug that is metabolized by DOPA decarboxylase to increase extracellular concentrations of norepinephrine in the brain[254]. The efficacy of droxidopa is predicted to be similar to the α2A-adrenoceptor agonists. Droxidopa is approved for adults with neurogenic orthostatic hypotension associated with primary autonomic failure (such as Parkinson's disease, multiple system atrophy or pure autonomic failure), dopamine β-hydroxylase deficiency or nondiabetic autonomic neuropathy ) and is administered orally[255]. Droxidopa is reported to significantly decrease the ADHD-RS-Total score[256]. Adverse events include headache, somnolence, depressed mood, suicidal ideation, myalgia, hyperhidrosis, insomnia, musculoskeletal stiffness, nausea, sedation, abnormal dreams and cough[256].

9.6.9.Baicalin

Scutellaria baicalensis Georgi (SBG), a Chinese traditional medicine, is used to treat fever, hepatitis, hypertension, inflammation, and jaundice[257]. Baicalin (or baicalein), a flavonoid purified from SBG, can protect against several neurotoxins which employ DAT[258,259]. Baicalin is found to decrease hyperactivity in the spontaneously hypertensive rat model and increase markers of striatal dopamine function. Therefore, it is proposed to be a potential medication for the treatment of ADHD[260].

Q4. Some anti-epileptic drugs have shown efficacy against ADHD, such as Lamotrigine. Hence, the Authors should incorporate such information, including the mechanism of action that may improve the novelty of this ms.  

R: Thank you for your excellent suggestion. We have added update information in our new MS.

Novel ASMs against both epilepsy and ADHD                                             

Lamotrigine (6-(2,3-dichlorophenyl)-1,2,4-triazine-3,5-diamine; molecular formula: C9H7N5Cl2; molecular weight: 256.09), developed by the GlaxoSmithKline, is a new generation broad-spectrum ASM for epilepsy and bipolar disorder. Lamotrigine was first approved by the US Food and Drug Administration (FDA) in 1994. Over the years, lamotrigine has been approved for the treatment of epilepsy and bipolar disorder[211] and guidelines have recommend this medication as first-line treatment for a monotherapy or polytherapy for focal and generalized seizures in both children and adults[212-216]. Mechanisms of the antiepileptic effect for lamotrigine include blocking voltage-gated sodium channels and voltage-gated calcium channels,  regulating the release of the inhibitory and excitatory neurotransmitters, stabilizing membrane potential of neurons, abolishing the repetitive firing of neurons [217-219].

Several studies show that lamotrigine can improve epilepsy without worsening ADHD symptoms[220-223]. It is not clear how lamotrigine can exert favorable effects on ADHD symptoms in CWE. Anti-epileptic effect of this medication may mainly contribute to this positive effect in the improvement of ADHD in CWE.

  1. The conclusion part should be revised; it is too long.

Thank you. The conclusion part has been largely revised.

The association between CWE and comorbid ADHD is still under-recognized in clinical practice despite the fact that the well-established bidirectional connection and shared genetic/non-genetic factors between epilepsy and comorbid ADHD largely rule out the possibility of a chance in this association. Stimulants are effective in children with comorbid ADHD, and the current body of evidence supports their safety in CWE within the approved dose. Early identification and proper management of comorbid ADHD are crucial to optimize the prognosis and reduce the risk of adverse long-term neurodevelopmental outcomes.

Reviewer 2 Report

The paper is well written. The scientific problem has been well introduced and the proposed method well explained and illustrated. however, some issues should be addressed. 

please add a list of abbreviations. 

please improve the introduction section, explain why your study is important.

Authors should do statistical analysis , and reports the results by charts, figures. 

improve the presentation, and remove any redundant information

Author Response

Reviewer B

The paper is well written. The scientific problem has been well introduced and the proposed method well explained and illustrated. however, some issues should be addressed. 

Q: please add a list of abbreviations. 

R: Thank you. We have added the following in the new version of the MS.

Abbreviations

ADHD          attention deficit hyperactivity disorder

AMP           amphetamine

ASC            amphetamine salt combination

ASMs          antiseizure medications

BECTS          benign epilepsy with centrotemporal spikes

BRE            benign rolandic epilepsy

CAE           childhood absence epilepsy

CWE           children with epilepsy

DAS           dextroamphetamine sulfate

DAT           dopamine transporters

D-MPH        dexmethylphenidate hydrochloride

DNRI          dopamine and norepinephrine reuptake inhibitor

DSM           Diagnostic and Statistical Manual of Mental Disorders

EEG           electroencephalogram

FDA           Food and Drug Administration

FLE           frontal lobe epilepsy

GABA         γ-aminobutyric acid

GWAS         genome-wide association studies

ILAE          International League Against Epilepsy

IQ             intelligence quotient

LDX           lisdexamfetamine dimesylate

MAH          methamphetamine hydrochloride

MECP2        methyl-CpG binding protein 2

MEF2C        myocyte enhancer factor 2

MPH          methylphenidate

MPHH       methylphenidate hydrochloride

NET           norepineph- rine transporter

SBG           Scutellaria baicalensis Georgi

SDX            Serdexmethylphenidate

SEEG          stereo electroencephalogram

SERT          serotonin transporter

SNP           single nucleotide polymorphism

VMAT2        vesicular monoamine transporter 2

VTX           Vortioxetine

Q: please improve the introduction section, explain why your study is important.

R:Thank you. The importance of this study has been addressed in the following.

Large population-based studies showed a significantly higher risk of ADHD in CWE than in the general population. The prevalence of ADHD was found to be as high as 77% in CWE, compared to 3-5% in the general population[13]. The presence of ADHD in CWE is challenging and can negatively affect the prognosis of the patients. Previous studies showed that CWE and concurrent ADHD are at increased risk of having an abnormal neurological examination and abnormal background rhythm on the electro-encephalogram (EEG)[14], lower ASMs adherence[15], and more negative outcomes in social, behavioral, and learning development compared with CWE alone[16].

Despite the high burden, comorbid ADHD remains unrecognized in the clinical setting. Besides, the current literature shows controversies regarding the magnitude and nature of ADHD in CWE, the connection between ADHD and ASMs, the challenges in managing epileptic children with ADHD, and the role of precision medicine in the management of ADHD in CWE. Thus, the current literature review aims to summarize the current burden of ADHD in childhood epilepsy, the possible mechanisms of association between epilepsy and ADHD, and the challenges imposed by ADHD during the management of childhood epilepsy.

R:Authors should do statistical analysis , and reports the results by charts, figures. 

improve the presentation, and remove any redundant information

 Q: Thank you. We have re-written the following sentences and deleted the redundant information and added a new table.

  1. 5. The burden of ADHD in CWE

The association between ADHD and CWE is under-recognized in clinical practice [42,43,78]. Statistical data shows that the reported prevalence of ADHD in CWE ranges from 12-70%, which is primarily attributed to disparity in screening tools, diagnostic criteria, and studied age groups[13,16,42,49,52,79-84] (Table 2). However, the prevalence of ADHD in healthy children without epilepsy is approximately 5-15%[42]. Obviously, CWE has a higher risk of ADHD than the general population.

Primarily inattentive presentation of ADHD appears to be more common than other presentations in CWE, which contradicts the distribution of ADHD presentations in the general population[85]. Previous reports showed that problems with attention were more common in ADHD patients with epilepsy than in children with ADHD alone[86]. However, other reports showed a similar distribution of ADHD presentations in children with ADHD and epilepsy and the general ADHD population[87].

Table 2.  Prevalence of ADHD in CWE

Country

published year

ADHD/CWE %

Reference

Brazil

2010

29.1

[82]

China

2012, 2018

24.7-42

[81,84].

Israel

2011

70

[79]

Iran

2016

60.4

[83]

Norway

2011, 2016

12.1-31.7

[42,49]

UK

2014, 2003

12-33

[78,80]

USA

2012

23

[52]

USA

2003

37.3

[13]

Regarding the statistical analysis, we would like to thank you for the foresight of our research. We do have applied for another research grant for your suggestion. We will publish our new results as soon as possible in the coming day. Thank you.

Q:     The conclusion part should be revised; it is too long.

R:  Thank you. The conclusion part has been largely revised.

The association between CWE and comorbid ADHD is still under-recognized in clinical practice, even though the well-established bidirectional connection and shared genet-ic/non-genetic factors between epilepsy and comorbid ADHD largely rule out the possibility of a chance in this association. Stimulants are effective in children with comorbid ADHD, and the current body of evidence supports their safety in CWE within the approved dose. Early identification and proper management of comorbid ADHD are cru-cial to optimize the prognosis and reduce the risk of adverse long-term neurodevel-opmental outcomes.

Round 2

Reviewer 2 Report

Authors have addressed my comments.